# Zoo-Led Initiatives and Their Role in Lemur Conservation In Situ

**DOI:** 10.3390/ani12202772

**Published:** 2022-10-14

**Authors:** Caterina Spiezio, Barbara Regaiolli, Margherita Savonitto, Simon Bruslund, Stefano Vaglio

**Affiliations:** 1Parco Natura Viva–Garda Zoological Park, 37012 Bussolengo, Italy; 2Fondazione ARCA, 37012 Bussolengo, Italy; 3Marlow Birdpark, 18337 Marlow, Germany; 4Conservation Committee of European Association of Zoos and Aquaria, 1018 CZ Amsterdam, The Netherlands; 5School of Life Sciences, University of Wolverhampton, Wolverhampton WV1 1LY, UK; 6University College—The Castle, Durham University, Durham DH1 3RW, UK

**Keywords:** lemur conservation projects, ex-situ and in-situ conservation, primate conservation, development cooperation projects, long-term studies, community engagement, European zoos, sustainable development goals

## Abstract

**Simple Summary:**

Lemurs are the most endangered group of primates. This preliminary study, based on data from the European Association of Zoos and Aquaria’s Conservation Database, aimed to gain new insights into lemur conservation by evaluating the role played by projects led by European zoos in Madagascar. We found that these conservation initiatives may have a positive impact on wild lemur conservation. However, we believe that improved communication on conservation efforts provided for a broader range of lemur species for each project would be needed to further engage the general public.

**Abstract:**

We examined wider society’s ability to achieve biodiversity conservation and management targets using lemurs as a case study. We evaluated the impact on lemur conservation in situ by conservation initiatives led by European zoos in Madagascar exploring the European Association of Zoos and Aquaria’s Conservation Database projects in terms of performed actions and achieved goals as well as communication to the public. We found that zoo-led conservation initiatives may have a positive impact on the ground and tend to achieve most conservation goals related to wild lemurs. However, we suggest that such conservation programmes should underline that they target further lemur species beyond the flagship charismatic species and that enhanced communication efforts would be beneficial to further raise public awareness.

## 1. Introduction

The Sustainable Development Goals (SDGs) were adopted in 2015 by the United Nations (UN) General Assembly and signed by the 193 member states of the UN, to promote global sustainable economic development by 2030 [1]. The 17 goals, which expanded upon the 8 Millennium Development Goals, are characterized by a cross-cutting and interdisciplinary nature [2]. A joint effort by all actors and stakeholders in society is needed to achieve the SDGs, but national governments are required to drive the process and take major responsibility as they are the signatory parties of the agreement.

Amid such actors and stakeholders, modern zoos assume a major role in ensuring the survival of endangered species via conservation, research, public education, and outreach activities [3,4]. They act as wildlife conservation organizations with projects around the world, working in partnership with local and national governments, civil society, and intergovernmental organizations (including the UN and its organizations), to design and implement conservation programmes. Particularly, according to the conservation strategy of the World Association of Zoos and Aquariums (WAZA) [5], zoos and aquaria must act both ex situ, by delivering the highest standards of care and welfare for their animals, and in situ, by providing safety for populations of species in the wild.

Among captive animals, zoo populations are unique as they are usually managed to educate the public regarding wildlife and their habitats (i.e., zoos can then help to meet SDG targets 15.1, 15.2, 15.3, 15.4 and 15.7, which are related to the areas of land under formal protection, sustainable forest management, fighting against deforestation, desertification, and the poaching and trafficking of protected species), and to preserve endangered species through captive breeding and reintroduction programmes (i.e., aligned with SDG target 15.5, which uses the International Union for Conservation of Nature (IUCN) Red List of Threatened Species to assess the risk of biodiversity extinction). In this context, the maintenance of the genetic diversity of such captive populations is imperative and these populations may serve as buffers against extinction and thus need to be viable for reintroduction into the wild.

Zoos are thus valuable assets for biodiversity conservation. They contribute to the conservation of endangered species through both ex situ breeding programmes and in situ reintroductions into the wild [6]. Captive populations held in zoological institutions worldwide act as a vital reservoir population for endangered species. First, they provide a safeguard against the extinction of species, with some species currently only existing in zoos. Secondly, in many cases they provide the opportunity for running conservation breeding programmes. These aim to preserve populations of species that are threatened with extinction in the wild; increase or maintain demographically stable numbers of individuals; maintain the genetic diversity and fitness of said populations; and, ultimately, reintroduce zoo-bred individuals into the wild either by establishing new populations or supplementing existing ones when deemed necessary [7].

Several institutions which are members of the European Association of Zoos and Aquaria (EAZA) are active in the conservation of habitats and entire ecosystems via conservation strategies that integrate in situ and ex situ management processes by delivering ex situ breeding programmes and supporting in situ projects. Research, conservation education, capacity building, advocacy, and fundraising are some of the actions necessary to work towards biodiversity conservation. In this context, cooperation, and communication, inside as well as outside of the zoo community, are crucial to achieve wildlife conservation goals [5]. Indeed, in situ conservation is about protecting animals but also improving the lives and health conditions of local human communities. Educating and empowering local communities to mitigate human–wildlife conflicts must be a priority in long-term conservation initiatives. To ensure the success of in situ conservation goals, sustainable development programmes with economic incentives focusing on livelihoods are necessary [5]. 

From a primate perspective, of the 504 species recognized today worldwide, almost half are classified as Endangered or Critically Endangered in the wild–primarily due to human impact; therefore, raising global scientific and public awareness of the plight of the world’s primates is now crucial [8,9].

Among all primate species, lemurs are the most endangered large taxonomic grouping. Almost a third of the 107 species currently living in Madagascar are listed as Critically Endangered, while 98% of them are threatened by extinction [10]. Lemurs are endemic to Madagascar, which is a recognized biodiversity hotspot and hosts around 21% of all primate genera and 37% of all primate families, making it the top priority in terms of primate conservation [11]. At the same time, Madagascar is also a growing eco-tourism destination, which implies potential for a positive shift towards biodiversity conservation (including protection and the reintroduction of endangered lemur species).

Outside of Madagascar, captive lemur populations are widespread in zoo collections worldwide, and many species are currently thriving in terms of abundance and demographic trend within EAZA member institutions [12]. Coordinated captive breeding programmes in zoological institutions, such as EAZA ex situ programmes (EEPs), can be important tools for conservation. Particularly, 14 of 107 lemur species are currently kept in EAZA institutions and managed via EEPs [13] as zoo-housed lemur populations and as such should serve to support their wild counterparts via zoo-led integrated conservation initiatives both ex situ and in situ [14]. 

EAZA zoos report the achieved goals of their conservation programmes via the EAZA Conservation Database showing their direct contribution to the conservation of biodiversity. This database is an online tool to keep track of conservation projects and to measure the EAZA conservation impact across the globe [15].

In this pilot work, we use lemurs as a case study to examine wider society’s ability to achieve the biodiversity conservation and management targets of SDG15 (Life on Land–“Protect, restore and promote sustainable use of terrestrial ecosystems, sustainably manage forests, combat desertification, and halt and reverse land degradation and halt biodiversity loss”), primarily with reference to SDG target 15.5 (“Take urgent and significant action to reduce the degradation of natural habitats, halt the loss of biodiversity and, by 2020, protect and prevent the extinction of threatened species”) and indicator 15.5.1, which uses the International Union for the Conservation of Nature (IUCN) Red List of Threatened Species to assess the risk of biodiversity extinction. Specifically, we explore the conservation initiatives led by European zoos and evaluate their achievements in relation to lemur conservation on the ground in Madagascar.

The main aim of this preliminary study is to evaluate the communication of measurable actions supported and implemented by European zoos with regard to the in situ conservation of lemurs by using zoo and conservation project websites as well as the EAZA Conservation Database.

## 2. Materials and Methods

### 2.1. EAZA Conservation Database

To select projects on lemur conservation we used the EAZA’s Conservation Database, an online tool [15] that helps to coordinate and keep track of conservation projects within the EAZA’s zoo community. All members can enter the database and update their yearly conservation contributions to specific projects. The achievement of the goals is therefore self-reported by each zoo. We decided to select only those projects with updated information of funding up to 2020/2022. In total 12 projects were selected.

For each project we recorded titles, target species, goals, duration, number of contributing zoos and number of dedicated websites. All projects selected for this study and their goals are listed in Table 1. All goals of different projects can be summarized as follows: support/educate local communities; habitat conservation; fundraising for species conservation; field research; capacity building; population monitoring; and rescue, rehabilitation, and reintroduction. These goals were considered in our analysis, focusing on whether they were performed or not, passing from goals to achievements.

### 2.2. Data Collection and Analysis

For each project, in addition to title, target species, duration and goals, we recorded the number of zoos involved and the number of webpages created by different zoos to communicate the project actions (Table 1); in contrast, for each project goal, we collected information on whether zoos effectively achieved them. We obtained this information through the EAZA’s Conservation Database. 

Zoo webpages were searched by projects to investigate whether zoos that filed information in the EAZA’s Conservation Database on each specific project also had information on their project webpage. Achieved goals were determined via the description written within the webpage dedicated to the specific project. The self-assessment of project achievements is an accepted practice, but we recognize that may not be fully objective. Communication and awareness raising are essential skills for conservation [16] and are relevant to create a relationship between conservation efforts and environmental education [17]. Webpages as means to communicate with the public can therefore be considered effective communication tools.

We investigated the communication and achievements of lemur conservation projects using non-parametric statistical tests, specifically Wilcoxon signed rank test and Spearman correlation. First, to test communication efficacy, we verified the relationship between the number of zoos involved in conservation projects and the number of webpages realized per project. To test the degree of efficiency of projects for in situ conservation we investigated the relationship between the number of goals and the number of those effectively achieved. We set significance level at *p* < 0.05 and analysed data using XLSTAT 2020.4.1.1023.

Step number 7 of the WAZA document on conservation strategy [5] states that developing a communication plan by the zoo has a positive and proactive value for visitors. In a digital world, webpages are a means to communicate and, therefore, we hypothesized that as a communication action the zoos would include conservation project information on their webpages.

## 3. Results

### 3.1. Project Communication

The number of zoos involved in lemur conservation projects was 84, whereas the number of webpages dedicated to such projects on the zoos’ websites was 42 (Figure 1) less than the zoos involved. The difference between the zoos involved in a project and the webpages of the zoo dedicated to the conservation project was significant (V = 45; *p* = 0.008) (Figure 2). However, we found significant correlation between the number of zoos involved in conservation projects and the number of webpages (Rho = 0.639; *p* = 0.029).

### 3.2. Goals and Achievements

Figure 3 reports the number of times different actions characterizing lemur conservation projects were set as goals and the number of times they were effectively achieved. Most of the projects had the following goals: “support/educate the local communities” and “habitat conservation”. We found a strongly significant positive correlation between goals and achievements (Rho = 0.954; *p* = 0.007).

### 3.3. Species and Conservation Projects

We found that six out of 12 conservation projects (50%) involved only one lemur species, three projects (25%) involved three lemur species, one project (8%) involved two lemur species, one project (8%) involved five lemur species and a further one (8%) had nine lemur species. Moreover, Figure 4 reports the number of projects involved in conservation (N = 24) for each species: 17 out of 24 of the species (71%) are involved in only one project, six out of 24 (25%) are involved in two projects and only one species (4%), *Varecia rubra*, is involved in three projects. Species that are involved in more than one initiative are *Cheirogaleus major*, *Daubentonia madagascariensis*, *Indri indri*, *Prolemur simus*, *Varecia rubra* and *Varecia variegata* (Table 1; Figure 4).

## 4. Discussion

Lemurs are the most endangered taxonomic group among primates [18]. Madagascar is considered one of the world’s foremost nature sanctuaries with its exceptional wealth of biodiversity, and unique flora and fauna. Unfortunately, particularly over the last two decades, anthropogenic change has drastically affected Madagascar’s natural resources and the forest has become degraded or vanished. Thousands of hectares of forest have been destroyed by logging and slash-and-burn agriculture. Many species of forest-dependent endemic Malagasy reptiles, birds, and mammals, including lemurs, are currently on the verge of extinction due to habitat loss and illegal exploitation [19]. 

In this scenario, developing reforestation programmes and supporting and educating local communities would be the key factors for successful conservation action; in addition, raising awareness about sustainable ecotourism as well as funds to support in situ conservation project are priorities agreed among conservationists world-wide [19]. In such a context, zoos can support, promote, and monitor actions such as implementing education in schools to achieve a new generation of Malagasy people aware of their native lemur species, working with Malagasy communities to ensure a sustainable use of the forest, and building capacity among local communities to protect lemurs and benefit from their conservation through ecotourism [20].

The aim of this study was to gain new insight into lemur conservation projects using the EAZA Conservation Database to evaluate the effects of initiatives led by European zoos.

First, we found that 24 lemur species were involved in 12 conservation projects carried out by 84 European zoos. Since 14 species are housed in EAZA members and belong to EEPs, we suggest that the lemur species collection in zoos should be prioritised according to species conservation needs [21,22]. However, Critically Endangered species, such as *Indri indri*, are not always successfully managed in zoos [22,23]. For these species, conservation goals could be achieved by hosting one model species and directing conservation efforts to protect another or multiple species [24]. 

Regarding communication of the projects, all but one project has at least one webpage describing aims and achievements. We found that the number of webpages for each project is significantly less than the number of zoos involved in different projects. However, the number of zoos is correlated with the number of webpages from their website. These findings suggest that each zoo involved in a lemur conservation project does not dedicate at least one webpage to communicate news of the project. However, we recognise that defining communication just via the presence of zoo webpages would be a very limited approach as the absence of a project webpage does not necessarily mean that a zoo is not communicating about the project itself. For instance, the zoo may have further dissemination activities on the project inside the zoo [20].

To add to this scenario, EAZA zoos are guided by EEP programmes that are linked to in situ conservation. The EEP programmes belong to Taxon Advisory Groups (TAG) which are active in defining actions for conservation programmes. With regard to lemurs, the EAZA Prosimian TAG has a well-developed “in situ conservation plan” and tools dedicated to communication such as their Newsletter. In addition, EAZA releases annual TAG reports in which research, education, and conservation achievements are described in detail. Nevertheless, overall, we believe that a greater effort should be undertaken by each zoo to enhance dissemination and public engagement about lemur conservation projects. In any case, more zoos certainly mean more communication initiatives [25,26]. 

Nowadays, to better communicate their efforts for conservation, zoos and aquaria must demonstrate that they are the conservation force claimed in their mission and vision statements. Each conservation project aims to implement lemur in situ conservation through different goals, from habitat preservation to fundraising for species conservation and encouraging local communities to participate in the efforts to preserve lemur habitat. We assessed whether the goals of different projects were aligned with the performed actions and achievements by involved European zoos. We found a strongly significant positive correlation between goals and achievements, suggesting that zoos can play an important role in the conservation of wild lemurs, not only on paper but also in reality, both ex situ and in situ [27,28,29]. Obviously, we recognise that such highly positive correlation between project goals and performed actions may be influenced by zoo self-reporting (i.e., zoos may be biased, or at least optimistic, about considering whether their goals have been met) and thus the interpretation of these data must be treated with precaution. However, we highlight that first we collected and analysed data about the communication of lemur conservation projects via zoo websites and then we considered the evidence of financial and staff-time investment by zoos into such projects via the EAZA Conservation Database. 

Impact-assessment methodologies have recently been developed to ensure that zoo efforts in the conservation of animals and their habitats are effective. Particularly, WAZA has designed the Project Conservation Impact Tool to provide a format to summarise project achievements and progress [30], whereas EAZA has developed the Conservation Database to collect all efforts provided by the EAZA member institutions in terms of actions, human resources, and financial investments through an online tool available to zoos for timely updates. Databases are crucial tools for an evidence-based approach to conservation, research, and education. This pilot work, focused on lemurs as a case study and based on the data collected through the EAZA Conservation Database, shows that there is potential to achieve sustainability targets in relation to biodiversity conservation and management under SDG15, especially with reference to SDG indicator 15.5.1, which uses the IUCN Red List Index to assess the risk of biodiversity extinction [31]. Conservation initiatives led by European zoos in Madagascar may impact lemur conservation via a number of actions aligned with SDG targets and indicators [32,33]. For instance, fundraising (SDG target 15a); capacity building (SDG target 15c); support/educate local communities (SDG target 15.2); habitat conservation (SDG targets 15.1, 15.2 and 15.4); field research (SDG target 15.5); population monitoring (SDG target 15.5); rescue, rehabilitation, and reintroduction (SDG target 15.5).

In many cases, the number of species involved in each conservation project ranges from one to three, whereas only two projects focus on more than three species of lemurs. In addition, most lemur species involved in conservation initiatives are part of one or two specific projects, whereas only one species, *Varecia rubra*, is involved in three projects (even though this is not the most common lemur species across zoos [34]). Nevertheless, the majority of projects have a wider approach and actually provide a kind of “added value” for a number of endangered species led by one “flagship” species. For instance, the project “*Helpsimus*” focuses on *Prolemur simus* but positively impacts other species living in the same area as well, such as *Eulemur rubriventer, Hapalemur ranomanfenisi, Avahi peyrierasi,* and likely further “overlooked species” [35]. Finally, as the primary goals to be reached are the education of local communities and habitat conservation, altogether these findings seem to suggest that one species could be used as a flagship species to achieve the goal of habitat conservation, satisfying the SDG indicators 15.1.2 and 15.4.1, which are related to land protection.

## 5. Conclusions

Based on the data from the EAZA Conservation Database, lemur conservation initiatives supported by European zoos seem to work effectively with regard to lemur conservation in situ, leading to the achievement of most project goals. This shows the importance of conservation strategies integrating in situ and ex situ actions. However, we suggest that the further improved communication of lemur conservation projects would be essential to raise awareness and funds, and thereby better support in situ conservation organised by zoos. With regard to lemur conservation planning, we recommend that strategies are carefully developed to address the current bias towards the involvement of only specific lemur species (with a broader range of species that could be targeted by zoo-led conservation efforts) and to ensure that project actions are well-aligned with UN SDG targets and indicators (including timing of their goals).

## Figures and Tables

**Figure 1 animals-12-02772-f001:**
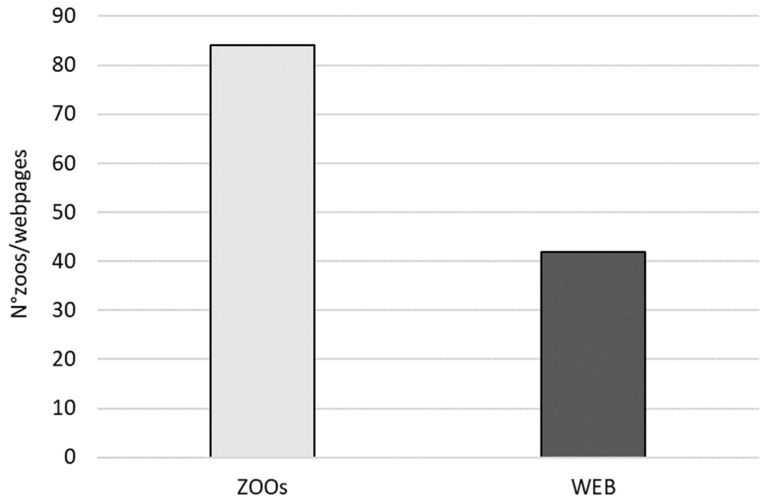
The number of zoos involved in all 12 conservation projects (ZOOs) and the number of webpages on websites of zoos involved (WEB).

**Figure 2 animals-12-02772-f002:**
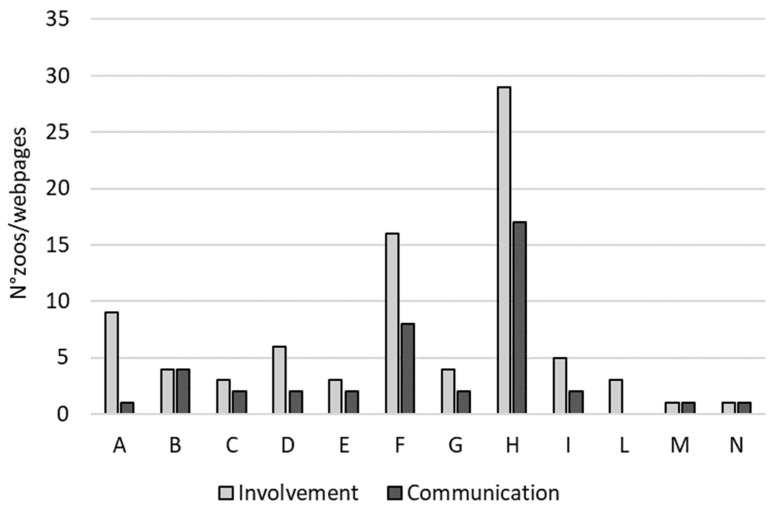
For each project (from A–N), the bar plot shows the number of zoos involved (Involvement) and the number of dedicated zoo-webpages (Communication).

**Figure 3 animals-12-02772-f003:**
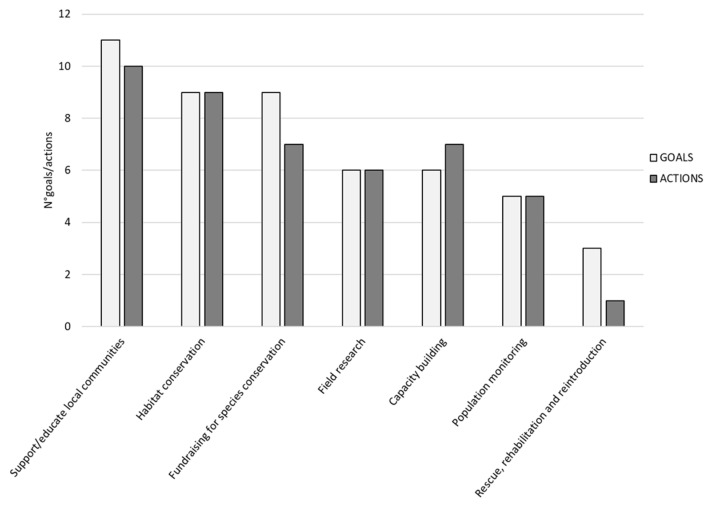
For all actions characterizing lemur conservation projects, the bar plot shows the number of times they were set as a goal and the number of times they were effectively achieved.

**Figure 4 animals-12-02772-f004:**
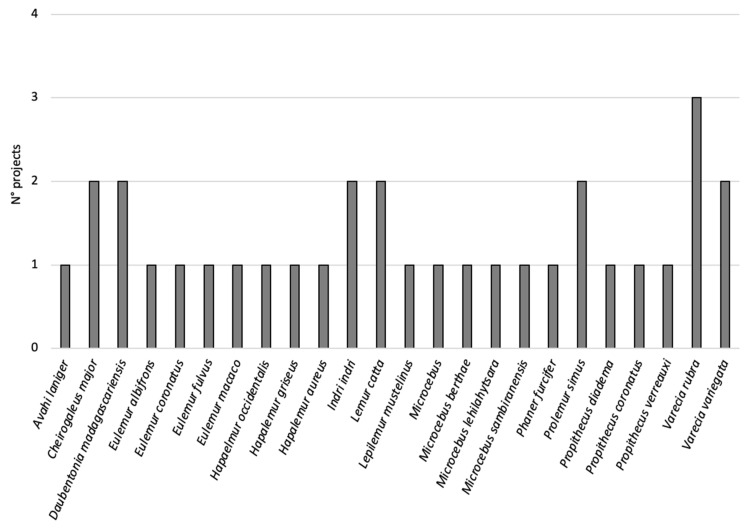
Lemur species involved in conservation projects.

**Table 1 animals-12-02772-t001:** Twelve lemur conservation projects used in the study. For each project the table reports name, species involved, goals, number of zoos involved (Zoos), and number of webpages on websites of zoos involved (WEB).

Project	Species	Goals	Zoos	WEB
A—Support to the Madagascar Fauna Group (MFG)	*Varecia rubra*	Habitat conservation, support/educate local communities, capacity building, fundraising for species conservation, sustainable agriculture.	9	1
B—Conservation of the Lemurs of Adriantantely Forest	*Hapalemur aureus, Varecia variegata, Indri indri*	Habitat conservation, field research, support/educate local communities, population monitoring, fundraising for species conservation.	4	4
C—Maromizaha Forest Conservation Project	*Hapalemur griseus, Lemur catta, Varecia variegata, Indri indri, Propithecus diadema*	Habitat conservation, field research, support/educate local communities, population monitoring, fundraising for species conservation, sustainable tourism.	3	2
D—Reniala Lemur Rescue Center	*Lemur catta*	Support/educate local communities, rescue, rehabilitation and reintroduction, fundraising for species conservation.	6	2
E—Red-ruffed Lemur Reintroduction Program	*Varecia rubra*	Rescue, rehabilitation and reintroduction.	3	2
F—Helpsimus Bamboo Lemur Programme	*Prolemur simus*	Habitat conservation, field research, support/educate local communities, population monitoring, capacity building, fundraising for species conservation.	16	8
G—Forest Conservation through Sustainable Development in Maroantsetra Region, Antongil Bay, Madagascar	*Varecia rubra, Hapaelmur occidentalis, Eulemur albifrons, Daubentonia madagascariensis, Cheirogalues major, Avahi laniger, Microcebus, Phaner furcifer, Hapalemur occidentalis*	Habitat conservation, field research, support/educate local communities, capacity building fundraising for species conservation, eco-tourism.	4	2
H—Support to AEECL—Association Européènne pour l’Etude et la Conservation des Lémuriens	*Eulemur coronatus, Eulemur macaco, Lepilemur mustelinus*	Habitat conservation, field research, support/educate local communities, population monitoring, fundraising for species conservation.	29	17
I—Sifaka Conservation Project	*Propithecus coronatus*	Habitat conservation, support/educate local communities, capacity building.	5	2
L—Chances for Nature in Madagascar	*Propithecus verreauxi, Microcebus berthae*	Habitat conservation, support/educate local communities.	3	0
M—Sustainable Management of Mangabe-Ranomena-Sahasarotra Protected Area	*Microcebus lehilahytsara, Cheirogaleus major, Daubentonia madagascariensis*	Habitat conservation, field research, support/educate local communities, population monitoring, capacity building, fundraising for species conservation.	1	1
N—Northern Madagascar	*Microcebus sambiranensis*	Field research, support/educate local communities, capacity building, fundraising for species conservation.	1	1

## Data Availability

The raw data supporting the conclusions of this article will be made available by the authors, without undue reservation.

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
