# Peer review of "Zoo-Led Initiatives and Their Role in Lemur Conservation In Situ"

_animals, 2022, doi:10.3390/ani12202772_

Round 1
Reviewer 1 Report
Line 21: ‘We examined the society’s ability….’ should read ‘We examined society’s ability….’
Format of figures - axis line needed, each axis needs a label, remove outline of figure so it looks like it is just part of the page and not in a box.
While this manuscript is very clearly written with minimal grammatical errors, the content do not match the title as impacts have not been systematically investigated.
Links to sustainability are minor and cannot be demonstrated - thus sustainability can be mentioned but should not be the focus of the Introduction.
The sample is very small and this study, even if a pilot, should be extended to a full audit of lemur conservation projects to have meaningful impact to the readership.
The methods should be expanded - for example how and when were zoo web-pages searched? How are 'achieved goals' determined? By the zoos? A project manager? How can existence of webpages by considered effective communication and engagement - there needs to be justification provided and consultation of the literature evidenced.
In the Discussion the highly positive correlation between goals and actions needs consideration - of course zoos will consider their goals met. How can this type of self-reporting influence outcomes and what precautions are needed when interpreting these data?
There is insufficient evidence to discuss one species kept in a zoo as an ambassador species - no data were collected to substantiate this conclusion.
Just because there is not a webpage on the project does not mean the zoo is not communicating about the project - the zoo may have interpretation on the project inside the zoo - this needs to be considered in the Discussion.
Reviewer 2 Report
There are many positives of this study. I like the association between SDG and zoo-led initiative, and this can be a great study to replicate in a larger scale. The paper is however missing some important contextual information. Before going into the specific of the study, you should provide a broader background on the importance of zoos for conservation, and link this to why zoos can help meeting SDGs. There are some information in there but I would expand that and change the order of your introduction.
Simple summary and abstract are missing important information. How you define effectiveness and positive impact should be clear from the simple summary and abstract. Missing a clear structure of the abstract, with introduction, aim, methods, results and discussion/conclusion.
You have up to 10 keywords you can use so I would add more keywords as your paper will be easier to find.
IUCN categories should be in upper case, e.g. Endangered
In the aim of the study and in general from introduction and method it should be clear how you assess efficacy and impact. A pilot study is something referred to field work, your study is just using a database
Not all of the actions listed in the methods are then showed in the results (e.g. anti-poaching, human-wildlife conflict)
Some of the results are not focused on your aim, or at least the link is not evident. Why do you expect the same number of zoos involved and webpages created? Why a correlation? In Fig 3 you put the main findings, but missing some goals? And how do they link to SDGs? Fig 4 is not so important, most of the info can be obtained from table 1. The species that are "protected" by more than one initiative can be listed in text as they are only 7
In general, it is not clear how the actions you selected are linked to SDG 15 and SDG 15 indicators and targets. Link 27 in the discussion is just to SDGs, not to targets used by the IUCN
Reviewer 3 Report
The authors attempted to address the Zoo initiative for lemur in situ conservation. But it needs specific issues to be addressed. Authors missed the World Zoo and Aquarium conservation strategies for lemur conservation to achieve the Sustainable Development Goals UN. A quantitative analysis is missing, the paper looks like a summarization of information. Strategies for the lemur conservation network must be addressed. The threats to lemur conservation must be addressed. The focus on conservation planning must be emphasised in the conclusion part.
Round 2
Reviewer 1 Report
I thank the authors for their extensive changes to this paper which I believe will be ready for publication following minor corrections for English and readability.
Author Response
We have now further amended the manuscript to improve its readability.
Reviewer 2 Report
I am happy with the new version of the paper. The paper is much improved and ready to be published.
Author Response
We have now further amended the manuscript with regard to English language and style, and so improved its readability.